# Cryopreservation of Semen in Domestic Animals: A Review of Current Challenges, Applications, and Prospective Strategies

**DOI:** 10.3390/ani12233271

**Published:** 2022-11-24

**Authors:** Mohsen Sharafi, Seyyed Mohsen Borghei-Rad, Maryam Hezavehei, Abdolhossein Shahverdi, James D. Benson

**Affiliations:** 1Department of Biology, University of Saskatchewan, Saskatoon, SK S7N 5E2, Canada; 2Semex Alliance, Guelph, ON N1H 6J2, Canada; 3Department of Embryology, Reproductive Biomedicine Research Center, Royan Institute for Reproductive Biomedicine, ACECR, Tehran 16635-148, Iran

**Keywords:** cryo-damage, sperm banking, sperm cryopreservation, antioxidant, membrane stabilization

## Abstract

**Simple Summary:**

Here we present a comprehensive review of the current literature describing mechanisms of damage incurred by cryopreservation, and we highlight several classical and novel strategies to mitigate this damage and increase sperm survival during the cryopreservation process.

**Abstract:**

Cryopreservation is a way to preserve germplasm with applications in agriculture, biotechnology, and conservation of endangered animals. Cryopreservation has been available for over a century, yet, using current methods, only around 50% of spermatozoa retain their viability after cryopreservation. This loss is associated with damage to different sperm components including the plasma membrane, nucleus, mitochondria, proteins, mRNAs, and microRNAs. To mitigate this damage, conventional strategies use chemical additives that include classical cryoprotectants such as glycerol, as well as antioxidants, fatty acids, sugars, amino acids, and membrane stabilizers. However, clearly current protocols do not prevent all damage. This may be due to the imperfect function of antioxidants and the probable conversion of media components to more toxic forms during cryopreservation.

## 1. Introduction

Mammalian sperm are among the first cells to be successfully cryopreserved, and over the last seven decades the use of cryopreserved semen for artificial insemination has come to play a crucial role in animal agriculture [1]. For most animal species, however, a large population of sperm are incapacitated after cooling to and warming from liquid nitrogen temperatures. Thus, to achieve equivalent artificial insemination fertility rates, several times more cryopreserved bull sperm are needed in comparison with fresh samples [2]. Although artificial insemination (AI) in pigs with frozen or fresh semen is common, the fertility of cooled semen remains about 50% of fresh semen [3]. After the insemination of superovulated ewes, pregnancy rates and fertility characteristics with frozen–thawed ram semen are reduced from fresh semen by 20% [4]. The long history of advancements in sperm cryopreservation in livestock has facilitated modern genetic strategies in breeding programs but has also been at the forefront in the assessment and mitigation of cryopreservation-induced damage, including osmotic, oxidative, and epigenetic damage that has expanded to somatic cell cryopreservation research. In this review, we first discuss a brief history and the importance of cryopreservation in livestock; we then highlight the biology, injuries, stresses, and challenges of sperm cryopreservation; and finally, we discuss conventional and modern strategies to prevent cryopreservation-induced damage to sperm. In short, this review aims to capture both classical and modern understanding of the modes and mitigation of damage induced by the cryopreservation process in domestic animal sperm.

It Is believed that the first successful AI was performed by Italian physiologist Lazzaro Spallanzani in 1780, who inseminated a dog using frozen–thawed semen [5]. Subsequent researchers repeated this work, including John Hunter in 1799, whose method resulted in human pregnancy [6]. At the same time in France, AI was shown to be a useful method of improving fertility in horses [7]. Following these preliminary reports, horse fertility centers began to spread throughout Europe for the collection and extension of semen and breeding of mares [7]. The various techniques for artificial insemination (AI), technician training, and revised methods for collection and cooling semen continued to increase, and by 1938 around 40,000 mares, 1.2 million cows, and 15 million sheep had been serviced by AI in Russia [8]. In the early 1800s the majority of the AI trials were being conducted in Europe and Russia; however, several successful efforts in horses and cattle had been reported in the USA as well [8]. For example, in 1907 a calf was produced via AI at the Oklahoma Experimental Station. After this achievement, the application of AI in the USA was developed, and in 1937 AI was applied in the cattle herds at the University of Missouri and subsequently at other universities [8].

Though references to the preservation of semen date back to the 1600s [9], it was not until the expansion of (AI) in the late 1950s and early 1960s—a time when the dairy cattle industry realized a need for longer storage of bull semen—that semen cryopreservation became a serious area of research. This continued study was founded on the basic discovery of Polge et al. (1949) that demonstrated that glycerol could enable the conservation of cells at low temperatures [10]. The progress in this front was rapid: the first offspring produced via cryopreserved semen was born in 1951 [11], and today the modern cattle industry is based on the use of AI with cryopreserved sperm. Table 1 shows the first recorded offspring produced via AI using frozen–thawed semen in different species. Beyond agriculture, human cryopreservation has provided flexibility in fertility options not only for couples but also for those undergoing iatrogenic treatments such as cancer therapy [12] or transgender women undergoing gender-affirming surgery [13]. 

## 2. Cryobiology of Sperm

Cryopreservation is the process by which cells or tissues can survive at low temperatures with reduced or ceased of metabolic activity [19]. During this process, however, cells experience various types of damage to all aspects of cell and tissue anatomy and physiology that consequently lead to reduced cell or tissue function [20]. In particular, semen cryopreservation leads to various physical, biochemical, and oxidative damages to the sperm membrane, leading decreases in the viability and fertilizing capacity of sperm [19]. These cryoinjuries mostly lead to loss of motility, plasma membrane functionality, and acrosome integrity of sperm [19]. Much of this damage occurs at the beginning and end of protocols when cryoprotective agents are added and removed [21], but damage also can happen during freezing and thawing at slow to moderate cooling and warming rates [22]. In a study by Bucak et al., several undesired changes in vacuole-like structure, as well as head, neck, acrosome, and mitochondria damages were observed via a scanning electron microscopy evaluation [23]. Moreover, freeze–thaw-induced damage to aquaporins in the plasma membrane have been found that could affect the plasma membrane’s functionality [24].

In the presence of extracellular ice during cooling, sperm experience very strong hypertonic conditions because the relative amount of water decreases in the extracellular unfrozen fraction [25]. This results in exosmosis to maintain an equilibrium between the intra- and extracellular solutes, leading to cellular dehydration, where the extent of this dehydration is generally inversely proportionate to cooling rate. The reverse process takes place during thawing and sperm face a relatively hypotonic condition resulting in swelling because of water uptake. In contrast, at fast cooling rates, classical cryobiology suggests that cellular dehydration is reduced and that intracellular ice formation is mainly responsible for cell death, a theory thoroughly documented in many somatic cell types [25]. However, sperm may be different: Morris et al. used cryo-scanning electron microscopy and freeze substitution to show that no intracellular ice is formed in sperm [26]. They claim that osmotic tension experienced during thawing is the reason for cell damage. To be clear, there is still debate about the presence of intracellular ice in sperm frozen at too-fast cooling rates. For example, the presence of intracellular ice is central to research by Devireddy et al., who use a calorimetry-based technique that works under the assumption that sperm lyse because of intracellular ice to identify the water permeability of sperm at subzero temperatures [27]. Clearly, more work in this area is justified. Nevertheless, the “two-factor hypothesis” describes two complementary modes of damage occurring at both ends of the cooling rate spectrum [28]: damage due to osmotic dehydration (known as solute effects injury for too-slow cooling protocols), and damage due to intracellular ice formation or, per Morris et al., osmotic imbalance during warming for too-fast cooling protocols [26]. This injury dichotomy suggests that there is an optimal cooling rate where damage is minimal. Membrane permeability to water (and its temperature dependence) is one of the main criteria determining the optimum cooling rate that itself depends on membrane composition as well as the presence of cryoprotective agents [27]. At subzero temperatures, a ”water transport model“ can be used to calculate cell volume changes during freezing and to predict optimal cooling rates that maximize dehydration (and thus minimize the likelihood of intracellular ice or osmotic imbalances) and minimize hypertonic exposure times [29]. 

## 3. Cryopreservation Injuries in Sperm

### 3.1. Changes in Sperm Plasma Membrane

The sperm plasma membrane is the major site of damage during cryopreservation (Figure 1). This damage can be divided into that which occurs during equilibration with cryoprotectant media and that which occurs during cooling [30]. As a high concentration (5–15% *v*/*v*) of permeating cryoprotective agents such as glycerol is almost always included in sperm cryopreservation media in most of the animal species, the equilibration process of sperm with these media can cause large changes in cell volume [22]. To wit, upon exposure to CPA containing media, sperm shrink and gradually reswell (see Figure 1 inset). Sperm are especially sensitive to these osmotically induced volume changes, where damage has been shown through loss of membrane integrity, loss of motility, and loss of acrosome integrity [31]. 

After CPA equilibration, sperm are cooled, which is another source of injury often seen at the plasma membrane [32]. First and foremost, the freeze–thaw process causes enormous changes in extracellular concentrations of the permeating CPA and other medium components. This freeze- and thaw-induced change in osmolality and the resulting osmosis can cause dramatic changes in cell volume, and thus significant mechanical stress on the cell membranes [33]. The freeze-concentration of the ionic components in the media are hypothesized to be a principle source of damage [34], and this damage is mitigated in some respect by the major nonionic media components, such as permeating (e.g., glycerol or ethylene glycol) or nonpermeating cryoprotectants (e.g., sucrose, raffinose, or trehalose), that reduce the relative ionic concentrations at subzero temperatures. Even in the absence of extracellular ice and concomitant large osmotic gradients, cooling-induced phase transitions change the ultrastructure of the plasma membrane [35]. In fact, most of the cryodamage in sperm is associated with the structural stability of the plasma membrane and is thus linked to plasma membrane composition [36]. For example, some phospholipids increase membrane flexibility, and cholesterol provides stability that seems to improve the resistance of sperm to freezing damage [37]. Membranes that have a low cholesterol to phospholipid ratio with an asymmetric pattern of cholesterol distribution seem to be more vulnerable to injuries [38]. Furthermore, lateral movement of membrane phospholipids is usually restricted in temperatures lower than 5 °C, and this ultimately results in a transition from the fluid to the gel phase. As a result of this phenomenon, membrane lipids are restructured, some cholesterol molecules are released, and many integral proteins in the plasma membrane such as ion channels become irreversibly clustered [39] in a way that can cause a loss of functionality [40]. All of these changes lead to destabilization of the membrane and a loss of its selective permeability, thereby increasing the influx of ions such as Ca^2+^ and bicarbonate from the extracellular space [19] that in itself can cause cryopreservation-induced sperm capacitation [41].

### 3.2. Changes in Sperm Plasma Membrane

The mechanisms responsible for DNA fragmentation after cryopreservation are still not completely understood, though it has been attributed to the increase in oxidative DNA damage [42]. Cryoinjuries on the sperm nucleus should be explored by distinguishing between different sites of injury, such as DNA and nucleoproteins. Sperm chromatin consists of DNA and protamine (P1 and P2) along with the histones that shape the nucleoprotein structure [43]. The distribution of protamine is different between species; for example, bull, boar, and ram have only P1 in the nucleoprotein structure while mouse, horse, and human present both P1 and P2. Additionally, P1 to P2 ratios are different between species, and this ratio affects the resistance of that structure to freeze–thawing procedures [44]: the extent of DNA-induced cryodamage is higher in species having both P1 and P2 rather than those exhibiting only P1 [44].

High DNA fragmentation and low mitochondrial membrane potential are significant contributors to reduced motility after cryopreservation [45]. The mitochondrial dysfunction that occurs following cryopreservation can result in the formation of reactive oxygen species (ROS). Interruption of oxidative phosphorylation and inactivation of the antioxidant enzymes are probably the main reasons for mitochondrial dysfunctionality [46]. Damages to the mitochondrial DNA and inner and outer mitochondrial membranes are the other reason for mitochondrial dysfunction. All these changes reduce the mitochondrial membrane potential, which can result in releasing free oxygen radicals through the membrane pores [47]. Mitochondria play a crucial role in ATP production by regulating oxidative phosphorylation and the tricarboxylic acid cycle important for the motility and fertilizing capacity of sperm [48].

### 3.3. Proteome Alterations

Studies have demonstrated that cryopreservation can alter the expression level of many proteins related to sperm function [49]. Moreover, the freezing process can cause protein degradation, phosphorylation, and carbonylation [50]. Recently, numerous comparisons of protein profiles have been reported between fresh and frozen sperm in several species to understand this aspect of cryoinjury (see [49] for review). The application of comparative proteomics has led to identifying specific proteins as potential key biomarkers for post-thaw recovery and sperm fertility in general [51]. For example, Chen et al. discovered that 41 proteins changed in boar spermatozoa during cryopreservation. These proteins were related to sperm motility, plasma membrane integrity, energy metabolism, capacitation, and sperm–oocyte fusion [52]. Expression of A-kinase anchoring protein (AKAP)-4, Fibrous sheath interacting protein 2 (FSIP2), Fascin, Ornithine decarboxylase antizyme 3 and Leucine-rich repeat, and coiled-coil centrosomal protein 1 are less present in frozen–thawed sperm [51]. On the other hand, the levels of fifteen proteins related to the reduction of fertility, including Nexin 1, Spermadhesins PSPI, Tetraspanin CD63 (CD63), Complement Factor D (CFD), and Ras GTPase-activating-like protein IQGAP2, were significantly increased after cryopreservation [51]. A recent study on boar sperm found that sperm membrane proteins such as Fc fragment of IgG binding protein, Lactadherin, Arylsulfatase a precursor and F-actin capping protein subunit alpha 1 are biomarkers that predict the likelihood of boar semen post-thaw functionality (also known as freezability) [53].

Similar to the results of studies conducted on boar sperm, proteomic analysis of ram sperm found significant changes in the abundance of 51 proteins. The proteins such as T-complex protein CCT subunits, Casein kinase I isoform gamma-2 isoform X2 (CSNK1G2), and TOM1-like protein 1 isoform X2 (TOM1L1) were decreased while the levels of other proteins such as Leukocyte elastase inhibitor (SERPINB1) and Tyrosine-protein kinase Fer isoform X2 (FER) were significantly increased in frozen sperm compared with fresh sperm [51]. He et al. also showed that cryopreservation decreased the expression of hexokinase1 (HXK1) and Casein kinase II subunit alpha (CSNK2A2) in ram sperm. They demonstrated that these proteins are in the sperm flagellum and thus may be associated with sperm motility and viability [48].

These cryopreservation-induced effects on proteins are now being explored in other species. For example, in rainbow trout sperm, cryopreservation can cause the release of proteins related to structure and metabolism from the mitochondria, cytoskeleton, nucleus, and cytosol to the extracellular fluid [54]. A recent study on Adria gazelle sperm found that 85 proteins differed between fresh and frozen–thawed samples. These proteins were mostly related to sperm metabolic pathways including mitochondrial energy production and glycolysis, which may explain the significant loss of motility in thawed sperm [55]. Ryu et al. found that cryopreservation success (freezability) biomarkers such as voltage-dependent anion-selective channel protein 2 (VDAC2) or glutathione s-transferase mu 5 (GSTM5) were different between bull sperm with high freezing tolerance index and low freezing tolerance index [56]. VDAC2 is a mitochondrial protein and plays an important role in the transportation of ions [57]. Therefore, alteration of this protein may induce ionic imbalance, which has adverse impacts on sperm function. Moreover, GSTM5 as a freezability biomarker plays a key role in regulating sperm resistance to oxidative stress [58]. Possibly relatedly, Gaitskell-Phillips et al. found that mitochondrial proteomics facilitated differentiation between good- and poor-freezing sperm in stallion [59], identifying six key proteins with more than a threefold change. In the chicken, Cheng et al. reported increases in 36 proteins and decreases in 19 proteins during cryopreservation. They found that proteins such as tubulin α-3, outer dense-fiber protein, and tektin5 were increased, while proteins including dynein and axonemal were decreased after freeze–thaw. Tubulin and Tektin are linked with flagellum structure and sperm motility [60], and flagellum-related protein alteration can decrease the motility of rooster sperm after cryopreservation. Furthermore, the activity of enzymes related to glycolysis, including glyceraldehyde-3-phosphate dehydrogenase and L-lactate dehydrogenase, was down-regulated after cryopreservation, which can cause a decrease in ATP content and motility in sperm [61]. In summary, the application of modern proteomics methods has yielded new insights into the full scope of changes and damage induced by sperm cryopreservation protocols.

### 3.4. Epigenetic Modifications

Epigenetics encompasses heritable mechanisms that include histone modifications, DNA methylation, and non-coding RNAs [62]. Epigenetic modifications could alter the performance of chromatin and help to regulate gene expression. Alterations in epigenetic patterns happen in response to environmental signals [63]. Recent studies demonstrated that increased oxidative stress in sperm cells during the freezing process resulted in epigenetic modifications and are the main reasons for the decrease in their motility and fertilization ability [63]. Importantly, these epigenetic changes could be transmitted to offspring and affect embryonic development [62]. Salehi et al. reported that epigenetics patterns such as DNA methylation (DNMT), histone methylation, and acetylation were reduced in rooster sperm after cryopreservation [64]. There is a negative correlation between free radicals, DNA fragmentation, and DNMT [65] that can reduce the fertilization potential of sperm after thawing.

Flores et al. found that protein-DNA disulfide bonds and histone H1-DNA binding proteins were altered in boar sperm after cryopreservation [66]. It is suggested that osmotic and oxidative tensions can cause an alteration in the structure of the nucleus during freezing and thawing [67]. Moreover, Aurich et al. stated that stallion sperm DNA cytosine methylation was increased by cryopreservation [68]. Another study reported that during cryopreservation, methylation levels increased in several genes, including CXCR4B, DND, POU5F1, VASA, SOX2, and SOX3 in zebrafish sperm, that may be correlated with the down-regulation of these genes [69]. A negative correlation has been reported between DNA fragmentation and DNA methylation that is most likely related to oxidative stress [65]. Increased methylation of some important genes such as POU5F1 and SOX2 in sperm can have adverse effects on early embryo development [70].

## 4. Prevention of Sperm Cryoinjury

Many methods are available for the cryopreservation of domestic animal sperm. While post-thaw viability varies from species to species and even specimen to specimen, current methods protect only about 50% of sperm against cryoinjury. To improve these results, numerous approaches have been tried to increase sperm resilience during the cryopreservation process. We organize them into two categories. First, we discuss conventional strategies that include cryoprotectants, antioxidants, membrane stabilizers, and other classical media modifications. Second, we introduce some novel recent approaches that have shown some promise. Figure 2 displays a summary of conventional and novel strategies that have been applied for semen cryopreservation in different animal species.

### 4.1. Conventional Strategies

During the last 25 years, the uses of various cryoprotectants and antioxidants have been the conventional strategies against cryoinjury that we will discuss in the following sections.

#### 4.1.1. Cryoprotectants

In general, semen extenders must have appropriate osmolarity and buffering capacity along with an adequate pH enabling them to protect sperm cells from cryogenic injury [69]. Generally, sperm cryopreservation extenders include one or both of two types of cryoprotectants: a non-permeating cryoprotectant (milk, egg yolk, soybean lecithin, or raffinose), or a penetrating cryoprotectant (glycerol or dimethyl sulfoxide (Me_2_SO)) [71].

Classical penetrating cryoprotective agents (CPAs) (e.g., glycerol and ethylene glycol) are known to provide a protective effect due to colligative properties [28]. At any given subzero temperature, the total unfrozen solution osmolality will be the same regardless of initial media constituents. In the absence of (relatively low toxicity) CPAs, the bulk of the osmolytes are salts. These salts cause significant damage during cooling and warming [72]. However, if the cryopreservation medium is supplemented with a significant percentage of penetrating CPA, then both the intra- and extracellular solutions retain the initial ratio of salt to CPA and at any given temperature, and the bulk of the osmolytes are CPA, not salt. This produces a less-damaging subzero environment for sperm. Classical CPAs also confer protection against damaging intracellular ice formation. However, as discussed above, Morris et al. demonstrated that this protection may be unnecessary in sperm, suggesting that the sperm intracellular water volume is too small to allow ice to nucleate and grow in the intracellular space [26]. A second benefit of penetrating cryoprotectants is that they improve the rearrangement of membrane lipids and proteins, membrane fluidity, and greater dehydration at lower temperatures that contribute to improved cryo-survival [38]. 

Although glycerol has many benefits as a CPA, there are some debates regarding its toxicity when it is applied in high concentrations [73]. Higher doses of glycerol are thought to be harmful in several aspects of cell function. Glycerol impacts plasma membrane coats such as glycocalyx, glycoproteins, and glycolipids and may deteriorate the membrane and increase the viscosity of cytosol [74]. Moreover, toxic levels of glycerol change the polymerization and depolymerization of α and β tubulins, the major proteins of the microtubules in sperm tail [73]. Si et al. assessed the motility and acrosome integrity of Rhesus monkey sperm that was frozen in the presence of different concentrations of glycerol (2%, 5%, 10%, and 15%), and they reported that the highest motility and highest acrosome integrity was achieved in 5% glycerol [75]. In the study by Bucak et al., with the use of trehalose in the extender as the modulator of glycerol toxicity, post-thaw quality indicators of ram sperm were significantly improved, and decreasing the glycerol content in the extender via supplementation with trehalose and taxifolin hydrate resulted in increasing the antioxidant capacity and reducing the oxidative stress [76].

On the other hand, a non-permeating cryoprotectant acts extracellularly without crossing the plasma membrane [77]. Therefore, non-permeating cryoprotectants such as sugars can provide similar colligative protection as their permeating counterparts, but only on the exterior of the cell. This, however, should be carefully considered: the advantage of permeating CPAs in the colligative sense is that they are associated only with a temporary change in water volume upon equilibration, whereas nonpermeating solutes such as sucrose or raffinose in similar molalities cause long-term and large-volume changes that are associated with membrane and cytoskeletal damage. Therefore, in general, species with tight volume tolerances (also known as osmotic tolerance limits) may benefit more from media containing permeating cryoprotectants over those that do not [77].

Egg yolk is the major component of extenders in most of the animal species and human and low-density lipoprotein is the effective fraction of egg yolk that protects sperm during freeze–thaw [78]. However, the challenges associated with using egg yolk that include microbial contamination have encouraged researchers to find a replacement [57]. Moreover, the wide variability of egg yolk compositions makes it difficult to standardize extenders, suggesting that sourcing egg yolk is an important part of any repeatable semen preservation protocol. It must be clear whether the eggs come from the same farm, are from the same breed or species, are being fed the same feed, etc. [79]. Because of the above drawbacks, several egg yolk alternatives have been explored in recent years. For example, soybean contains a high component of low-density lipoproteins such as lecithin, which is a plant-based extender ingredient. Soybean lecithin has recently been successfully used as an extracellular cryoprotectant for cryopreservation of semen in ram [80], goat [81], bull [82], buffalo [83], and human [84]. This non-animal origin protectant has improved the viability, mitochondrial membrane potential, and acrosome integrity in ram sperm [78] and reduced the lipid peroxidation of goat semen [81].

#### 4.1.2. Antioxidants

Sperm contains a high amount of polyunsaturated fatty acid (PUFA) that makes it prone to lipid peroxidation due to the massive production of reactive oxygen species (ROS) during cryopreservation [49]. Sperm regulate and react to this oxidative stress through various intrinsic antioxidant protective systems that exist in both sperm and seminal plasma including superoxide dismutase (SOD), catalase (CAT), glutathione peroxidase (GPx), and glutathione reductase (GR), as well as non-enzymatic antioxidants such as methionine, vitamin C (ascorbic acid), and vitamin E (α-tocopherol) [85]. The capacity, however, is not enough [86], and the performance of these antioxidants could be reduced by dilution or cooling, resulting in decreased benefits of the endogenous antioxidative defense. Thus, as a cryopreservation strategy, antioxidants are used to reduce the detrimental effect of ROS on sperm during cryopreservation [85].

The two main categories of antioxidants are enzymatic and nonenzymatic and both have been well documented as effective strategies for improving cryorecovery and mitigating cryoinjury. The enzyme superoxide dismutase (SOD) in the cytoplasm (Cu, Zn—SOD) and the mitochondria (Mn-SOD) is responsible for combining with two frequent ROS molecules including the superoxide anion ($O_{2}^{-}$) and hydrogen peroxide [85]. However, there are controversial results regarding the beneficial effects of SOD. While the addition of SOD to extenders provided higher-motility sperm during cooling and freezing preservation [87,88], Silva et al. did not find any increase in sperm kinematic parameters after adding SOD to the ram semen extender [89]. Nevertheless, acrosome integrity and mitochondrial activity were improved in the presence of SOD [89]. Roca et al. reported that catalase improves the fertilizing potential of boar sperm after thawing by using catalase alone or in combination with SOD [90]. In addition, Fernandez-Santos et al. demonstrated that catalase inhibits DNA damages during the oxidative stress of cryopreservation [91]. We also can find numerous studies describing the beneficial effects of glutathione for semen cryopreservation in broad ranges of species. In fresh semen, glutathione did not affect the quality of ram sperm [89]. In another study, Silva et al. demonstrated that glutathione (2 mM and 5 mM) preserved the acrosome integrity of ram sperm [89]. Moreover, 5 mM oxidized glutathione increased the motion and velocity characteristics of ram sperm after freeze–thaw [92], and similar results were found at lower concentrations in turkey sperm [93]. Similar results have been found using the multifunctional antioxidant melatonin, where positive effects for sperm cryopreservation were found in pig [94], ram [95], goat [96], canine [97], fish [98], bovine [99], and human [100]. 

Alpha tocopherol (vitamin E) is one of the major compounds with antioxidative properties that is frequently found in the plasma membrane and seminal plasma. This lipophilic antioxidant protects fatty acid contents of membranes against peroxidation [86] and it has a dose-dependent effect [101]. The trolox analogue of vitamin E, which is soluble in water, improved several quality indicators of boar semen during cooling storage [102]. In addition, it increased fertilizing capacity and reduced the amount of hydrogen peroxide in bull sperm when it was added to the extender [103]. The beneficial effects of vitamin E on reproductive capacity have also been shown in chicken [104], boar [105], rabbit [106], ram [107], and buck [108] when added as supplementation to their diets.

Ascorbic acid (vitamin C) is another water-soluble vitamin attributed with reproduction; however, its exact mechanism is still uncertain [109]. Vitamin C might have a significant effect on the protection of DNA during cryopreservation [110]. Azawi and Hussein reported an improvement in the motility and viability of ram sperm supplemented with 0.9 mg/mL vitamin C during preservation at room temperature [111]. In a contradictory report, vitamin C reduced the motility of ram sperm when extenders were enriched with 50 mM or 100 mM compared to the control group [112]. Vitamin C might be a pro-oxidative compound in the presence of ferrous ions in the extender [85], as it converts Fe^3+^ into Fe^2+^, resulting in a reaction with oxygen or hydrogen peroxide, which then triggers lipid peroxidation [113].

Amino acids are available in seminal plasma and they are counted as non-enzymatic scavengers with antioxidant characteristics. Various types of amino acids such as hypotaurine, glutamine, cysteine, taurine, histidine, proline, and glycine were found to reduce DNA fragmentation and enhance the various post-thaw parameters of ram sperm [114]. Sangeeta et al. found that the addition of 25 mM l-proline and 20 mM l-glutamine in a Tris-based medium decreased lipid peroxidation and enhanced the acrosome integrity of sperm [115]. Fattah et al. found that the addition of 1 mM and 2 mM l-carnitine in cryopreservation media improved the mitochondrial function of sperm and resulted in higher progressive motility after thawing [116]. However, combinations of these amino acids may result in a negative impact on the semen quality, as Zhandi and Sharafi found that combining cysteine and glutathione in soybean-lecithin-based extender increased apoptosis in their post-thaw analysis of ram sperm [117]. Another amino acid that is frequently used in the various types of extenders is bovine serum albumin (BSA), which can protect the membrane integrity of sperm, especially during heat stress [118]. In some trials, 10% or 15% BSA have been attempted as a substitute for egg yolk in ram semen diluents and these demonstrated an equal cryoprotective effect compared with the egg yolk [119,120]. Finally, in a study by Coyan et al., the addition of 1, 2, and 4 mM ergothioneine reduced the percentage of DNA fragmentation in post-thaw ram sperm [121]. Ergothioneine is a low-molecular-mass thiol that is present in some tissues. It scavenges oxygen hydroxyl radicals and peroxyl radicals and acts as a regulator of iron metabolism, and it has been shown that ergothioneine protects sperm from oxidative stress and improved the post-thaw motility of ram [121] and canine [122] sperm.

#### 4.1.3. Sugars

Sugars play multifunctional roles not only for regulating osmotic pressure and reducing relative ion concentrations, but also for stabilizing proteins and phospholipid bilayers during the freezing process [123]. They also have been considered not only as a source of energy for the sperm during cryopreservation but also as a means to prevent damage to different structural, sub-structural, and biochemical organs [124]. Trehalose is a disaccharide formed by binding two D-glucose molecules and is one the most effective sugars for cryopreservation [125]. Trehalose increases the distance between the membrane phospholipids by binding to the polar portion of the phospholipids in the plasma membrane. This distance may inhibit the formation of ice in the plasma membrane by helping the flow of water molecules from the cell and consequently stabilize the plasma membranes. Moreover, trehalose protects the transmembrane ion channels, reducing water leakage both into and out of the cells [126], and has been associated with modifications of the proteins during cryopreservation of ram sperm [127]. While trehalose is a popular nonpermeating cryoprotectant sugar alone or in combination with other additives [128], there has been much success in sperm cryopreservation media recipes that use other types of sugars, most notably in mouse sperm, which relies nearly entirely on the inclusion of raffinose [129]. This being said, some sugars that have positive effects in one species (e.g., raffinose in mouse) have negative effects in others (e.g., raffinose in chicken [130]). 

Lactose is another sugar additive that is mainly used in pig semen freezing media [131]. Lactose can modulate the tonicity of solutions as well as interact with main groups of membrane phospholipids, and thus could increase the stability of the membrane during the freeze–thaw process [132]. Freezing media supplemented with lactose and trehalose improved quality of swine sperm post-thawing [133]. Chanapiwat et al. observed that lactose increases recovery of boar sperm compared to other sugars such as fructose, glucose, and sorbitol when they used them in an egg yolk-based media [134]. In another trial, replacement of lactose by trehalose and sucrose reduced oxidative stress markers and improved the quality of the frozen–thawed sperm [132].

A brief survey of the types of sugars used in animal semen cryopreservation extender are shown in Table 2.

#### 4.1.4. Membrane Stabilizers

During cooling, phospholipids in the membrane undergo a phase transition from a liquid phase to the crystalline gel phase. Integral proteins are excluded from the crystalline gel domains and the membrane becomes unstable [152]. Additionally, cooling-induced efflux of cholesterol from the membrane induces capacitation-like changes in frozen sperm [152]. Several articles have reported that the treatment of sperm before cryopreservation with cyclodextrins loaded with cholesterol leads to improved plasma membrane integrity and increases the osmotic tolerance of sperm [153,154,155].

There are several reports on improved survival rates after the inclusion of cholesterol during cryopreservation of various species’ sperm [156,157]. For instance, the addition of cholesterol-loaded-cyclodextrin (CLC; also referred to as cholesterol-cyclodextrin complexes) in boar sperm improved plasma membrane and acrosome integrity and decreased lipid peroxidation after cryopreservation [158]. CLC increases the membrane and acrosomal integrity of bull sperm following cryopreservation [159]. Khellouf et al. found cholesterol and vitamin E both preloaded in cyclodextrins can improve protection in frozen bovine sperm against cold shock and oxidative stress [160]. Chuaychu-Noo et al. reported that treatment of rooster sperm with CLC before cryopreservation improved the quality of frozen sperm and fertility rate [161]. Murphy et al. found that pretreating stallion sperm with cholesterol before freezing reduced superoxide generation and increased post-thaw sperm viability [162]. Wojtusik et al. demonstrated that adding CLC to addra gazelle sperm prior to freezing prevented the loss of metabolism and motility-associated proteins such as CAPZB, HS90A, and PGAM2, as well as improving post-thaw sperm motility [55]. Adding CLC in skim-milk-based extender enhanced cryoresistance in ram sperm and improved sperm motility, sperm plasma membrane integrity, and fertility [153]. However, CLC and other cholesterol products are not a panacea: adding cholesterol-loaded cyclodextrin before cryopreservation did not increase fertility in sheep [163], horses [164], and donkeys [165], and exogenous cholesterol incorporated into the sperm membrane using cholesterol-loaded cyclodextrin can impair capacitation-related mechanisms, reducing fertility in frozen–thawed sperm [156]. 

#### 4.1.5. Using Dietary Additives

As the impact of membrane lipid constituents affects post-thaw viability, it is reasonable to expect that pre-ejaculate diet may impact semen quality, if not cryopreservation success. Towards this, it has been reported that diet components and formulation affect the semen quality in the animal. Zanussi et al. demonstrated that dietary 2% flaxseed oil plus 200 mg/kg vitamin E for 60 days improved aged broiler breeder rooster semen parameters and reproductive performance. This was attributed to possibly increased testosterone, reduced lipid peroxidation, and modification in the content of DPA, DHA, and arachidonic acid of the plasma membrane [166]. In horses, researchers found that complementing stallions’ diet with DHA not only increased the DHA in sperm but also increased the kinetic parameters of cryopreserved sperm [167]. Ansari et al. reported that oral usage of D-aspartic acid increased the concentration of sperm in the rooster and improved the fertility and hatchability rates in hens inseminated with those sperm [168]. Generally, the impact of diet on sperm cryosurvival seems to be marginal, but diet modifications in composition, feeding regimens, and trial length could affect sperm freezability [169]. In other studies, fish oil (3% dry matter of diet) significantly improved different post-thaw ram sperm parameters, while palm and sunflower oil (3% dry matter of diet) neither enhanced nor negatively affected ram semen characteristics [170,171]. In another report, dietary consumption of oleic acid in rams resulted in higher total antioxidant capacity and superoxide dismutase and lowered the amounts of malondialdehyde during liquid storage of semen [172]. 

#### 4.1.6. Warming/Thawing

There are significant interactions among cooling rate, warming rate, and survival in cryopreserved semen [173]. From a fundamental cryobiology point of view, samples that are cooled quickly must be warmed quickly, because samples that are cooled in a quasi equilibrium or nonequilibrium manner will be subject to deleterious recrystallization upon warming, so warming rates must be sufficient to outrun this process [174]. Semen is no exception [175]. Hernández found that more-rapid warming resulted in increased motion parameters in boar sperm, with significant interactions among warming rates and glycerol concentrations [176]. In the bull, Lysachenko found that cryopreserved sperm had better motility when warmed more rapidly in temperatures up to 70 °C [177], extending studies by Correa et al. who only warmed at temperatures up to 37 °C [178]. However, rapid warming rates are more difficult to achieve. Hernández et al., and subsequently Tomás-Almenar and de Mercado, used a 70 °C water bath to achieve this and noted sensitivity on the scale of seconds between successful warming and killing sperm [175,176]. 

### 4.2. Novel Strategies

Classical approaches to improve recovery after cryopreservation have been mostly chemical, with some careful attention to adjusting cooling and warming rates. Here we highlight several novel approaches outside of these paradigms that have been applied to increase sperm cryosurvival. Some of these approaches include mild sublethal stress induced before freezing, induction of a magnetic field before freezing, nanoparticle-enhanced cryopreservation, sperm preservation via freeze-drying, and monolayer centrifugation.

#### 4.2.1. Induction of Mild Sublethal Stress before Freezing

Mild sublethal stresses induced before freezing, including hydrostatic [179], osmotic [180], or oxidative stress [181], increase sperm resistance against cryo-injury. All the above-mentioned studies intended to induce responses that can prevent the apoptotic pathway during cryopreservation. For example, Huang et al. stated that the induction of very low hydrostatic pressure before cryopreservation enhanced the ubiquinol-cytochrome C reductase complex protein that is thought to play a critical role in regulating sperm motility [179]. This protein also participates in various cellular functions, such as cell cycle control, protein stabilization, redox regulation, fatty acid metabolism, and scavenging damaged proteins [182]. In another study, using a very low concentration of nitric oxide (NO) as a free radical that can induce mild oxidative stress significantly reduced the caspase-3 activity in cryopreserved bull sperm [181]. Similarly, Hezavehei et al. reported that the moderate stress generated by using 0.01 μM nitric oxide in human sperm resulted in increased post-thaw motility and viability and and reduced apoptosis-like changes after cryopreservation [183]. Furthermore, they found that the effect of sublethal nitrosative stress on human sperm is a time-dependent response [183]. Their results demonstrated that the application of sublethal nitrosative stress (0.01 μM) before the cryopreservation process for 60 min could protect sperm against cryo-injuries. Another interesting sublethal stress recently applied to semen cryopreservation is photobiomodulation, where sublethal laser light was applied to human [184], turkey [185], bovine [186], and rabbit [187] sperm with improved post-thaw recovery. While the mechanism of success is not well understood, Safian et al. found reduced levels of ROS and lipid peroxidation in the laser-irradiated pretreatment groups compared to controls [184].

#### 4.2.2. Induction of a Magnetic Field before Freezing

The use of “magnetized” extenders treated with a high magnetic field has increased the cryo-survival of sperm in some species. This approach has been employed to interrupt the regular network of water molecules in the cryopreservation medium to reduce the formation of hexagonal ice crystal, which is the most destructive form of ice for sperm. [188,189]. Via the application of a magnetic field during extender preparation, it is thought that the size of the ice crystal could be smaller and might enhance sperm survival after cryopreservation. Askarianzadeh et al. observed a higher quality of rooster sperm when the freezing media were prepared under a 2000–4000 gauss magnetic field [190]. The magnetized extender also decreased ROS concentrations in post-thaw boar sperm [190]. In a recent study, cryopreserved human sperm was improved by exposing the freezing diluents to a 1000 Hz ELEF [191].

#### 4.2.3. Nanoparticle-Enhanced Cryopreservation Media

As their name suggests, nanoparticles (NPs) are particles at the nanometer scale, with flexible fabrication and a high surface area to volume ratio [192]. In the reproductive system, some nanoparticles and nanomaterials have been demonstrated to improve fertility outcomes [193]. Recently, studies reported that the application of NPs with antioxidative properties has considerable capability to improve the results of cryopreservation protocols (Table 2) [194,195,196]. Falchi et al. have shown that cerium oxide (CeO_2_) nanoparticles have beneficial effects on ram spermatozoa during cooling [192]. They found that the cerium oxide (CeO_2_) NPs act as ROS scavengers to protect the integrity of DNA and plasma membranes. 

It was reported that selenium nanoparticles (SeNPs) have been used as a scavenger of ROS to protect sperm cells against oxidative stress [197]. Safa et al. demonstrated that the addition of vitamin E combined with SeNPs into rooster semen extender could increase the total antioxidant capacity (TAC) level and reduce malondialdehyde (MDA) concentration, compared to a control extender after freezing [197]. Adding SeNPs to semen extender improved bull sperm quality and the in vivo fertility rates after the freeze–thawing process [196]. Moreover, Mehdipour et al. found that lecithin nanoliposome extender could decline the percentage of apoptotic sperm and improved the quality of ram sperm after thawing [198]. Recently, Nadri et al. investigated the effect of vitamin E and glutathione peroxidase (GPx) in a lecithin-nanoliposome-based extender for bull sperm quality after cryopreservation [195]. They found that sperm viability and blastocyst formation rate were increased in nano-lecithin-based extender with 1.0 mM GPx [195]. 

#### 4.2.4. Sperm Preservation via Freeze-Drying (Lyophilization)

The freeze-drying of sperm is a simple preservation technology that has been investigated for decades but has sparked much recent enquiry. It is an alternative, low-cost storage option for biodiversity preservation of domestic species [199]. An attractive feature of freeze-drying is that after drying sperm can be stored at 4 °C and transported at room temperature, requiring no liquid nitrogen [200]. In theory, their small size and low water volume suggest that sperm may be good candidates for successful freezing-drying [201]. At the least as a preservation or conservation tactic, even nonviable freeze-dried sperm have intact chromatin and can be used for fertilization via intracytoplasmic sperm injection [202]. Wakayama and Yanagimachi reported that freeze-dried mouse spermatozoa were able to create normal embryos and generate normal offspring, although the sperm were motionless and appeared dead [203].

Moreover, other studies reported live offspring in mice [204], rat [200], rabbit [205], and horse [206] via fertilization with lyophilized sperm. Restrepo et al. compared three cryopreservation methods including freezing, vitrification, and freeze-drying on the quality parameters of equine spermatozoa; they found that freeze-drying produced a decline in lipid peroxidation (LPO) and enhanced mitochondrial membrane potential (lΔΨM) when compared to other methods [46].

#### 4.2.5. Monolayer Centrifugation

Single-layer centrifugation (SLC) has been shown to separate robust sperm from the rest of the ejaculates; therefore, it can increase the percentage of cells with higher resistance against cryo-damages. This approach, which is based on colloid centrifugation, can select sperm with desirable characteristics and consequently could end up with more viable sperm post-thawing [207]. In this approach, sperm is layered over a colloid and gently centrifuged, then robust sperm move through the colloid to form a pellet, while seminal plasma is retained at the top and cells with lower quality or reduced cryo-resistance are retained at the interface between the sample and colloid [208]. In previous studies, sperm were selected with better chromatin integrity and a higher proportion of high mitochondrial membrane potential [208]. This approach can potentially select the good quality semen that might not reach the minimum cryopreservation threshold for freezing. It should be noted that SLC has produced varied results depending on the density of colloid used in trials; lowering the density of the colloid tends to reduce its selective capacity [209].

## 5. Sperm Cryobanking Expansion in Research and Industry

We are in the midst of a biodiversity crisis that has been termed the sixth mass extinction event in the history of Earth [210]. Loss of genetic diversity of livestock and disease outbreak due to limited genetic resources is threatening the current production farming system. One way to address this issue is through the technology of cryopreservation: the genetics of many local or endangered breeds/species with dwindling or valuable genetic pools may be stored under the form of cryopreserved reproductive cells [211]. Beyond the frequently considered wild species conservation efforts, domesticated animal genetic resource conservation is an important challenge to maintain domestic biodiversity and adaptation of animal species to global changes or breeding accidents or epidemics [212]. Reproductive cell cryopreservation also represents security for farm animal genetics and may be a useful tool to contribute to the measurement of genetic progress.

Biobanking is a rapidly growing industry, covering diverse fields such as human medicine, farm animal production, laboratory animal record-keeping, and wildlife. Different reproductive cell types are now stored in cryobanks: mainly semen and embryos in mammalian species, semen and primordial germ cells in birds, semen and germ cells in fish, and semen and larvae in shellfish. Some cryobanks also include somatic cells with the hope that, in the future, these cells will be reprogrammed to become inefficient reproductive cells [213]. In France, the National Cryobank of Domestic Animals contains reproductive cells and somatic tissues of avian species, mammals, fish, or shellfish conserved as semen, embryos, or larvae depending on the species. This progress in reproductive physiology and biotechnologies has made possible the extension of the range of species available in cryobanks.

## 6. Conclusions and Future Perspective

The cryopreservation of domestic animal sperm is a complex procedure that involves the regulation of many agents to achieve desirable results. To ensure even minimal success, suitable diluents, sperm dilution rates, cooling rates, and thawing rates are needed. Moreover, complete knowledge of the complex sperm physiology for each species is required to maximize the post-thaw recovery of sperm and consequently, fertility.

Although the first successful sperm cryopreservation was over 60 years ago, current methods result in post-thaw spermatozoa with about 50% post-thaw viability for most species. Likely due to the wide variation among species’ sperm, a standardized cryopreservation process has not been established. Some measures, including specific cryoprotectants, have been applied to improve the survival and fertility of frozen animal sperm. Factors leading to spermatozoa cryoinjury are complicated and have not been fully explored, and the extensive and severe cryoinjury to the structure and physiological function of domestic animal sperm have been confirmed. Currently, the cryoinjury mechanisms in mammalian and avian spermatozoa are of considerable interest in the field of cryobiology and reproductive science. Because the functional role of spermatozoa is regulated by proteins and nucleic acids, a molecular explanation may be necessary. Research related to cryopreservation of domestic animal semen has progressed greatly in recent years; however, significant room for improvement still exists. The current technological advances, such as vitrification, freeze-drying, and high-throughput sequencing technologies, can provide a new perspective to improve the cryopreservation efficiency of animal sperm. 

## Figures and Tables

**Figure 1 animals-12-03271-f001:**
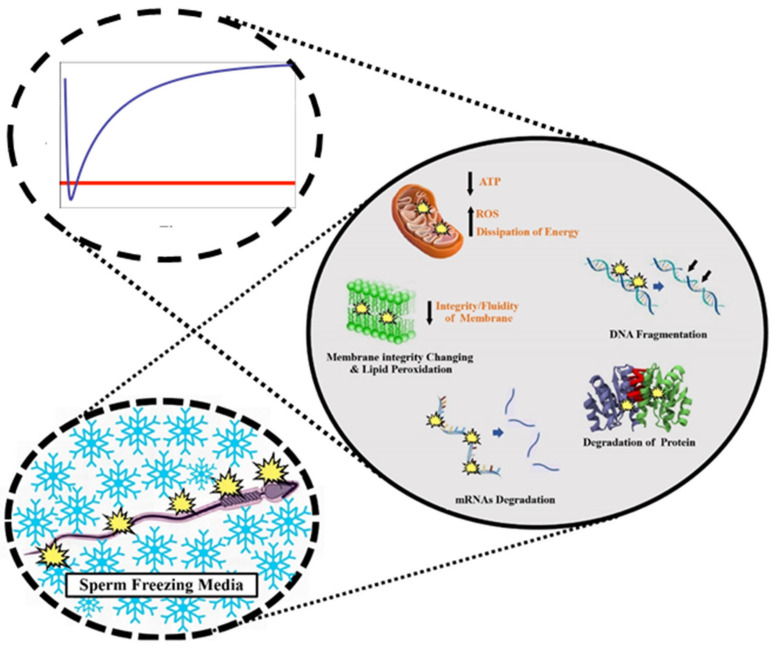
Mechanisms and foci of cryopreservation-induced injury (**right**) in sperm under two regimes: equilibration with and from high concentrations of cryoprotectant media which induces potentially damaging volume excursions (blue line representing volume, and red line representing an osmotic tolerance limit; **top left**) and cooling in the presence and absence of extracellular ice (**bottom left**).

**Figure 2 animals-12-03271-f002:**
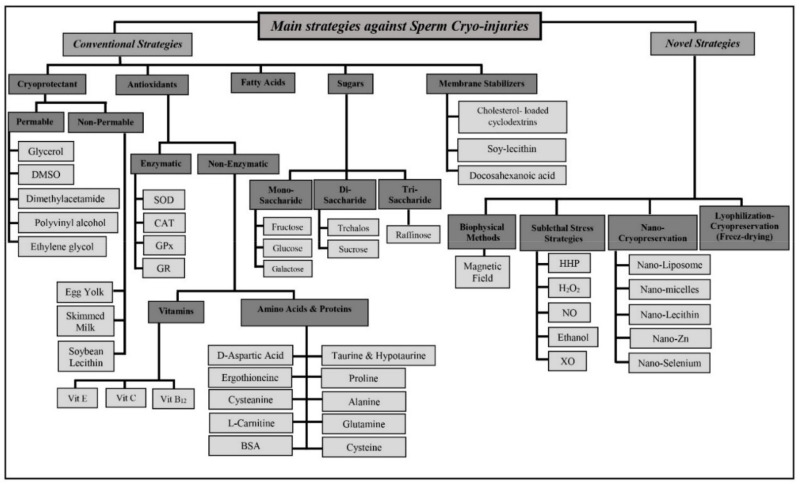
Main strategies to prevent animal sperm cryo-injuries (note all acronyms are in the abbreviation list).

**Table 1 animals-12-03271-t001:** First recorded offspring produced via AI using frozen–thawed semen in different species.

Species	Year	Reference
Avian	1942	[14]
Bovine	1951	[11]
Human	1953	[15]
Porcine	1957	[16]
Equine	1957	[17]
Ovine	1967	[18]

**Table 2 animals-12-03271-t002:** Example literature for several sugar additives used in animal semen extenders.

Sugars	Species	Results	Authors
Trehalose	Ram	Increased motility of frozen–thawed sperm	[135]
		Improved motility of frozen–thawed sperm	[136]
		Improved viability and membrane integrity of frozen–thawed sperm	[137]
		Improved post-thaw parameters	[126]
		Improved acrosome integrity	[138]
		Improved kinetic parameters, morphology, membrane integrity, and mitochondrial activity	[139]
		Improved post-thaw recovery using 50 and 100 mM trehalose with slow cooling	[140]
		Increased post-thaw parameters using combination of 3% glycerol and 60 mM trehalose	[141]
		Improved survival rate during cold storage using combination of 50 mM taurine and 50 mM trehalose	[142]
		Improved ultrastructural morphology of sperm using combination of 1.5% ethylene glycol and 100 mM trehalose	[143]
	Goat	Increased post-thaw motility and acrosome integrity	[124]
	Buffalo	Improved motility, viability, and membrane integrity	[144]
	Bull	Improved post-thaw mitochondrial activity and viability	[145]
Fructose	Boar	Fructose-based extender improved post-thaw motility and viability	[134]
	Bull	Fructose-based extender improved motility, plasma membrane integrity on the 3rd, 5th, and 7th day of storage	[146]
Glucose	Ram	Glucose improved post-thaw parameters	[147]
	Boar	Glucose increased post-thaw recovery	[148]
Raffinose	Ram	Improved motility, viability, mitochondrial activity of frozen–thawed sperm	[149]
		Increased viability and motility and decreased acrosome abnormalities	[137]
	Chicken	Reduced fertility performance	[130]
	Bull	Improved motility and plasma membrane integrity on 3rd, 5th and 7th day of storage	[146]
	Mouse	Raffinose is required for the standard inbred mouse sperm cryopreservation protocol	[150]
Sucrose	Bull	Improved motility, acrosome integrity, and plasma membrane functionality	[151]
	Stallion	Increased several kinetic parameters using 100 mM sucrose	[152]

## Data Availability

Not applicable.

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
