# Peer review of "Cryopreservation of Semen in Domestic Animals: A Review of Current Challenges, Applications, and Prospective Strategies"

_animals, 2022, doi:10.3390/ani12233271_

Round 1

Reviewer 1 Report

In this paper, Sharafi et al., focused their attention on an issue Cryopreservation of semen in domestic animals.

They focus on different points: Cryobiology of Sperm, Cryopreservation injuries in sperm, Prevention of sperm cryoinjury, and Sperm cryobanking expansion in research and industry.

The Prevention of sperm cryoinjury point, having to have information on all domestic species, is somewhat concise, but it gives a few hints of where research is being done to prevent damage from freezing.

The review is interesting and well conducted, above all as it investigates a very important issue like cryopreservation and now is very poor investigated; I recommend for its publication in Animals prior these minor revisions:

-          Format:

Line 94, 438, 549, there is one more space after the point

Figure 2: Missing capital letter in the name of Polyvinyl alcohol

Although it is not mandatory, in a review it would be very important to put the DOI of all the references used.

-          In sugars, lactose is not mentioned. This sugar used mainly in the cryopreservation of boar semen, must be included and referenced.

-          Glucose is also used in pigs; a citation should be included in Table 2.

-          A point should be placed in point 4.1 commenting on the thawing process, since it is a fundamental aspect in cryopreservation processes, where advances and improvements are also made.

Reviewer 2 Report

Dear Editor and Authors,

I consider the topic of the article of interest for researchers and academics in the field of animal sciences, more precisely animal reproduction and biology. The approach and structure of the paper are appropriate, the vocabulary used in the manuscript is varied and the transition between phrases is smooth, making the text easily understandable. There are some minor adjustments that must be made with the English proofreading, but overall the manuscript seems suitable. I suggest modifying some phrases that seemed a little more difficult to understand.

This phrase is difficult to understand, please try to reformulate “For  many  animals,  however, many sperm are infertile after cooling and warming to liquid nitrogen temperatures. For example,  to  achieve equivalent  fertility  rates,  several  times  more  cryopreserved  bull sperm  must  be used  compared  with  fresh  samples”.

In fact, much of the cryodamage in sperm acts on the structural stability of the plasma membrane [36]. This damage is associated with plasma membrane composition

I suggest “Such is the case of boar semen…” instead of “Like  the  boar” ….proteomic  analysis  of  ram  sperm  showed  significant  changes  in  the abundance of  proteins.”

I suggest taking into consideration the introduction of a paragraph regarding the use of single-layer centrifugation as an approach for reducing cryodamage and improving the cryo survival of semen. There are many articles and information regarding this matter. The method has now been used successfully in the conservation of semen from endangered species, which is why I consider that readers would surely benefit from the additional information provided in this article.

Round 2

Reviewer 2 Report

The authors have significantly improved the quality and readiness of the manuscript.